# Using GOES-R ABI Full-Disk Reflectance as a Calibration Source for the GOES Imager Visible Channels

Andrew K. Heidinger [1,*], Michael J. Foster [2], Kenneth R. Knapp [3] and Timothy J. Schmit [4]

1 GeoXO Program Office, NOAA/NESDIS, Greenbelt, MD 20771, USA
2 Cooperative Institute for Meteorological Satellite Studies (CIMSS), Madison, WI 53706, USA; mike.foster@ssec.wisc.edu
3 National Centers for Environmental Information (NCEI), NOAA/NESDIS, Asheville, NC 28801, USA; ken.knapp@noaa.gov
4 Center for Satellite Applications and Research (STAR), NOAA/NESDIS, Madison, WI 53706, USA; tim.j.schmit@noaa.gov
* Correspondence: andrew.heidinger@noaa.gov

**Abstract:** The availability of onboard calibration for solar reflectance channels on recently launched advanced geostationary imagers provides an opportunity to revisit the calibration of the visible channels on past geostationary imagers, which lacked onboard calibration systems. This study used the data from the Advanced Baseline Imager (ABI) on GOES-16 and GOES-17 to calibrate the visible channels on the GOES-IP (GOES-8, -9, -10, -11, -12, -13, and -15) sensors (1994–2021). The visible channels are dominant sources of information for many of the essential climate variables from these sensors. The technique developed uses the stability of the integrated full-disk reflectance to define a calibration target that is applied to past sensors to generate new calibration equations. These equations are found to be stable and agree well with other established techniques. Given the lack of assumptions and ease of application, this technique offers a new calibration method that can be used to complement existing techniques used by the operational space agencies with the GSICS Project. In addition, its simplicity allows for its application to data that existed prior to many of the reference data employed in current calibration methods.

**Keywords:** GOES; calibration; climate

## 1. Introduction

The earth has been observed by geostationary meteorological satellites nearly continuously since the mid-1970s. All of these satellites flew imagers with visible reflectance channels that were not calibrated on board the satellite and have relied on vicarious post-launch calibration activities. With the recent launch of the advanced geostationary imagers by JMA, NOAA, and KMA, the historical geostationary imagers are being replaced with imagers that have onboard calibration systems for the solar reflectance channels [1]. The goal of this study was to explore the utility of leveraging the calibration of the new sensors to calibrate the visible channels of the older sensors and whether the integrated full-disk visible imagery is stable enough to serve as a calibration target.

The calibration of visible channels on geostationary and polar-orbiting satellite imagers which lack onboard calibration has been an active topic for many years. Techniques have been developed which leverage the onboard calibrated solar reflectance channels on polar orbiting satellites [2], use of well characterized and stable earth targets such as deserts and snow sheets [3–6], and the use of the stable reflectance of deep convective clouds [7]. Some of the techniques have advanced to the point of being implemented by space agencies with the Global Space-based Inter-Calibration System (GSICS) [8]. However, the use of integrated full-disk reflectance as a calibration target has not yet been explored. The calibration used in the International Satellite Cloud Climatology Project (ISCCP) did assume that reflectance of

the earth's surface was a stable target [9,10], and Reference [11] used metrics of the full-disk reflectance distribution to calibrate the Meteosat First Generation Imager visible channel. The philosophy of GSICS is to promote multiple and varied calibration methods to achieve robust results. If this method proves worthy, its applicability to GSICS could be pursued.

Our target in this study was the use of the Advanced Baseline Imager (ABI) data [12,13] on GOES-16 and GOES-17 to recalibrate the visible channel on the GOES-8/15 imagers (GOES-8, -9, -10, -11, -12, -13, and -15). At the time of the writing of this paper, GOES-15 had been partially retired; GOES-14 was still in on-orbit storage and its data record was not yet sufficient for inclusion in this study; and GOES-13 had moved over the Indian Ocean. Table 1 provides the launch dates of the GOES satellites that are used in the calibration equations.

**Table 1.** Dates used in the calibration of GOES-IP Satellites.

| Satellite | Time (Years) | | | | |
|---|---|---|---|---|---|
| | Launch | Operational | Calibration Start | First Valid Data | Last Valid Data |
| GOES-8 | 1994.28 | 1994.83 | 1995.44 | 1995.17 | 2003.25 |
| GOES-9 | 1995.39 | 1995.74 | N/A | 1996.05 | 1998.55 |
| GOES-10 | 1997.31 | 1998.51 | 2000.00 | 1998.64 | 2006.47 |
| GOES-11 | 2000.33 | 2006.47 | 2006.47 | 2006.49 | 2011.93 |
| GOES-12 | 2001.56 | 2003.25 | 2003.25 | 2003.30 | 2010.28 |
| GOES-13 | 2006.39 | 2010.28 | 2010.28 | 2010.31 | 2018.02 |
| GOES-14 | 2009.48 | N/A | N/A | N/A | N/A |
| GOES-15 | 2010.17 | 2011.93 | 2011.65 | 2011.95 | 2020.0 |
| GOES-16 | 2016.88 | | | | |
| GOES-17 | 2018.16 | | | | |

In addition to pursuing this technique to complement the existing techniques used in GSICS, the GOES-IP calibration information derived here will be used in future versions of the Pathfinder Atmospheres Extended (PATMOS-x) cloud climate data records which currently exist on the NOAA Advanced Very High-Resolution Radiometer (AVHRR) data record (1979–2020) [14]. The calibration results will therefore be referred to as PATMOS-x when comparing to other methods and data. The technique developed is also motivated by plans to extend the International Satellite Cloud Climatology Project (ISCCP) to the advanced geostationary imagers being launched recently and in the coming years. Tying the future and past geostationary imager records together in a radiometric context will be critical for the success of this effort.

## 2. Data

The GOES-IP data were obtained from the Space Science and Engineering Center (SSEC) at the University of Wisconsin–Madison. The GOES-IP files used are McIDAS AREA formatted files [15], and the GOES-R files are original Level-1b files that were provided by NOAA and archived at SSEC. All processing was accomplished by using the Clouds from AVHRR Extended (CLAVR-x) processing system [14]. Both GOES-IP and GOES-R Level-1b data provide data at mixed spatial resolutions. CLAVR-x operates only at the spatial resolution of the thermal channels. Both the GOES-IP and GOES-R 0.65 micrometer (visible) channels have smaller footprint resolutions, providing 16 visible pixels within the area of each thermal pixel. To generate 0.65 micrometer data at the spatial resolution of thermal channels, the data are sampled where the value of the upper left corner is selected for both GOES-IP and GOES-R. Additionally, GOES-IP pixels overlap neighboring pixels by about 50% in the east–west (along scan) direction [16]. For this study, we removed every other thermal pixel in the along-scan direction for the GOES-IP data. For GOES-IP, this resulted in approximately 7 million pixels, and for GOES-R, this resulted in approximately 29 million pixels for every 0.65 micrometer full-disk image. Sensitivity studies of further

sampling the GOES-R data to match the number and size of the GOES-IP data did not show any impact on the integrated full-disk quantities used here.

Note that, for the GOES-R data, the nominal calibration was applied, and this includes the large adjustment (~7%) made in April 2019 [17] to the visible channel (band-2). This adjustment was also included in the data before this adjustment was made. The root cause of this bias in the GOES-R ABI visible channel is still unknown.

## 3. Methodology

As stated above, the use of stable targets as a calibration source is not new. Specific targets located on the earth, however, require many steps to be useful. They require cloud-clearing, angular modeling, and atmospheric correction. All of these steps potentially add biases and noise to any derived calibration information. Methods that compare geostationary to low earth-orbiting sensors require precise angular and temporal matching of the two data. The method proposed here avoids all of these steps by using the entire sun-lit full disk. The justification for this is that the earth is largely in radiative balance between the incoming solar, the outgoing solar, and the outgoing longwave radiation. The outgoing solar is determined by the planetary albedo, assuming that the incoming solar is constant. An analysis of the planetary albedo by radiation budget satellites, such as the Cloud's and the Earth's Radiant Energy System (CERES) (see References [18,19]), shows the planetary albedo to be stable over the last two decades. Any one geostationary imager sees roughly 1/6 of the earth's surface. If the stability of the planetary albedo holds for the smaller regions seen by any one geostationary satellite, the albedo over full disk offers the opportunity of a stable calibration target that requires no detailed processing.

This paper applies this method to GOES-East and GOES-West data from NOAA. GOES-East is located at 75W, and GOES-West is located at approximately 135W. The GOES-IP data measured the full disk every 3 h. This method assumes that full-disk reflectance is stable when measured at the same time and at the same location. For this study, the noontime full-disk data were used. For GOES-East, this scan started at 17:45 UTC, and for GOES-West, this scan started at 21:00 UTC. During the time that GOES-16 was in the GOES-East position, its scans started at 17:45 UTC and then switched to 17:50 UTC. GOES-16 is located at 75.2W, and GOES-17 is located at 137.2W. It is assumed that these slight departures in space and time are not important. GOES-16 started observing at the GOES-East position in December 2017, and GOES-17 started observing at the GOES-West position in November 2018. At the writing of this paper, GOES-16 had been in the GOES-East position for roughly four years, and GOES-17 had been in the GOES-West position for roughly three years. Throughout this analysis, the full disk is defined as the portion of the disk illuminated with solar zenith angles less than 80 degrees. Within this region, 85% of the pixels must have valid reflectance to be included in the analysis.

The traditional quantity used in calibration is scaled radiance, *R*, which is defined as follows:

$$R(\mu, \varphi) = 100\pi\rho^2 I(\mu, \varphi) \Big/ \overline{F}_o, \tag{1}$$

where *I* is the radiance, $\mu$ is cosine of the viewing zenith angle, $\varphi$ is the viewing azimuth angle, $\rho$ is the sun–earth distance factor (defined later), and $\overline{F}_o$ is the annual-mean in-band solar irradiance. The parameter we use as the stable calibration reference is the full disk scaled radiance, $R_{fd}$, which is approximated in this analysis as follows:

$$R_{fd} = \overline{R(\mu, \varphi)}, \tag{2}$$

which is simply the mean of *R* for each pixel in the illuminated portion of the full disk. We tried weighting the mean calculations by $\mu$ and by the cosine of the latitude and found little impact on the resulting calibration slopes; therefore, we used the simplest method. The weighting techniques have little impact since they are applied to both the scaled radiance and to the observed counts, and it is the ratio of these two quantities that matters for computing the calibration.

Figures 1 and 2 show the monthly mean images of near-noon scaled radiance for GOES-16 and GOES-17, respectively. It is interesting to note that the images show coherent structures over the multiple years included in the averages such as the annual variations in the Intertropical Convergence Zone (ITZC) and stratocumulus fields off the coasts of North and South America. The presence of snow in winter in North America is not a distinctive feature in these images.

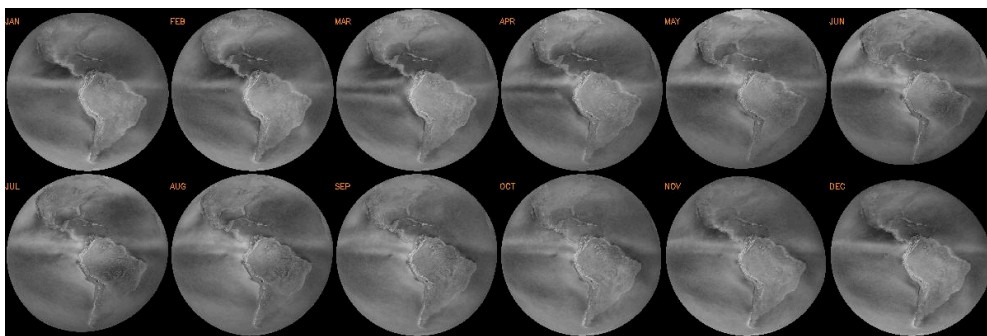

**Figure 1.** Mean monthly scaled radiance images for GOES-16 (GOES-East) at 17:45 and 17:50 UTC. Range is a linear scale from 0 to 100%. Solar and viewing zenith angles are limited to 80 degrees. Data range is from December 2017 through April 2022.

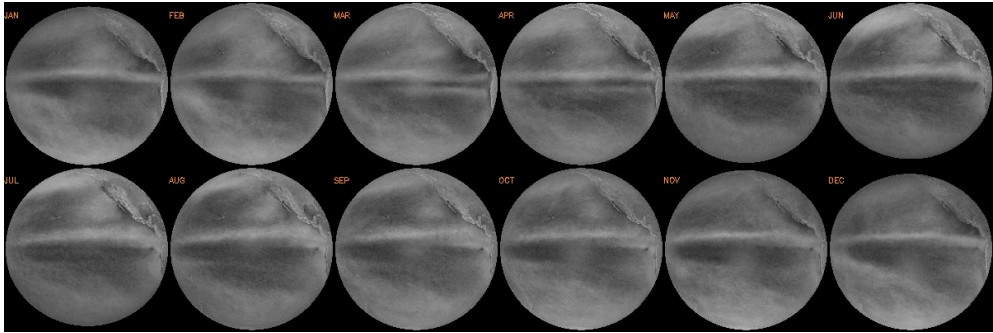

**Figure 2.** Mean monthly scaled radiance images for GOES-17 (GOES-West) at 21:00 UTC. Data range is from January 2019 through April 2022.

Figure 3 shows the annual cycle in the monthly mean $R_{fd}$ for GOES-16 and GOES-17 for the period when they were in the operational GOES-East and GOES-West positions. The range for GOES-16 is 18.5 to 20.5%, and the range for GOES-17 is 17 to 19%. The annual cycle shows a semiannual cycle, with peaks occurring around the vernal and autumnal equinoxes. Figures 3 and 4 also show the mean and standard deviation of the annual cycles in $R_{fd}$ derived later from the GOES-IP data. Because of the small number of years of GOES-R data to generate the reference curves, the standard deviation of the GOES-R reference is ignored and the values from the 20+ years of GOES-IP are assumed to be a truer estimate of the standard deviation of the annual cycle of $R_{fd}$. The values from GOES-IP observations are used to define the uncertainties when deriving the calibration curves. The standard deviations from the shorter period of the GOES-R reference data are larger than the standard deviation of the longer period GOES-IP data. The observed GOES-IP standard deviation of the monthly values of $R_{fd}$ are generally less than 1%, except for the winter months of GOES-West, which have values near and slightly above 1%. As will be described, the small values of the standard deviation in the monthly values of $R_{fd}$ and the small biases between the $R_{fd}$ curves from GOES-R and GOES-IP indicate that the $R_{fd}$ metric is a suitable basis for using as a calibration reference. Table 2 lists these monthly mean and standard deviation values used to the calibration slopes in the next section.

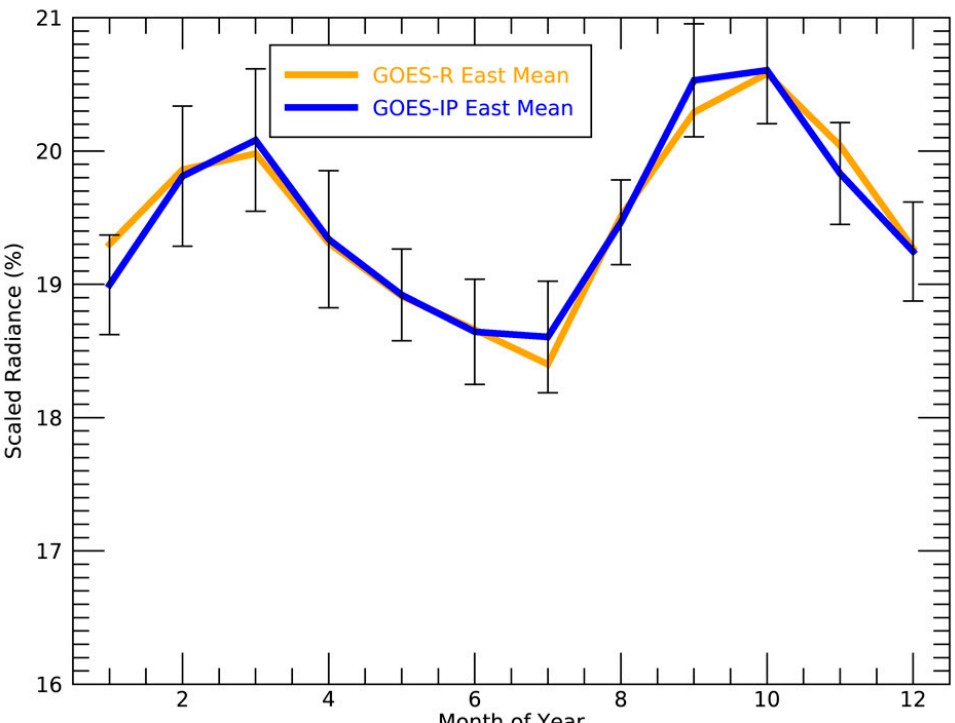

**Figure 3.** The annual cycle of FD scaled radiances for GOES-East. The orange curve is the annual cycle for GOES-R (GOES-16) and is used as the reference values for calibrating GOES-IP East data. The blue curve is the resulting annual cycle from the calibrated GOES-IP East data. The error bars show the standard deviation of the monthly values for the entire GOES-IP record. These standard deviations are assumed to be representative of the GOES-R reference.

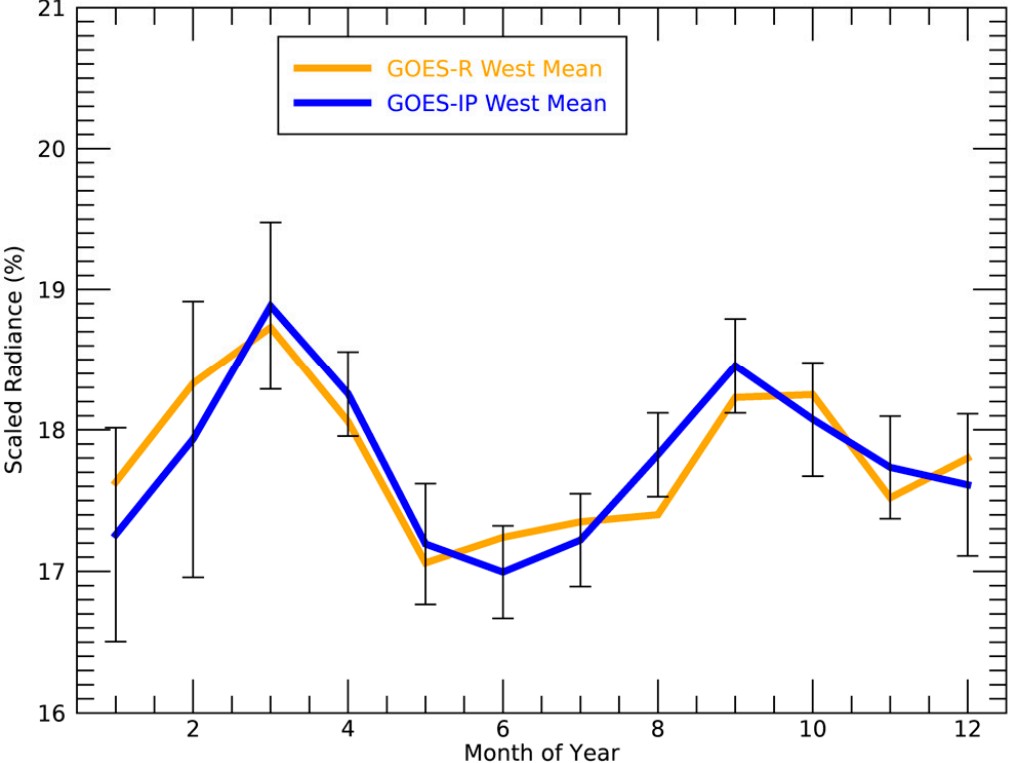

**Figure 4.** Same as Figure 3 but for GOES-West, where GOES-R reference is GOES-17.

**Table 2.** Monthly means and standard deviations (SD) of the full-disk scaled radiances for GOES-West and GOES-East used as the reference values and computed from GOES-R. The observed standard deviations computed from the GOES-IP series that are used in the fitting process are also shown.

| | GOES-West Reference Mean (%) | GOES-West Reference SD (%) | GOES-West Observed SD (%) | GOES-East Reference Mean (%) | GOES-East Reference SD (%) | GOES-East Observed SD (%) |
|---|---|---|---|---|---|---|
| January | 18.2 | 0.96 | 0.76 | 19.2 | 0.78 | 0.37 |
| February | 19.0 | 1.2 | 0.98 | 19.7 | 0.80 | 0.53 |
| March | 19.3 | 0.84 | 0.59 | 19.9 | 0.81 | 0.53 |
| April | 18.8 | 0.99 | 0.30 | 19.3 | 0.77 | 0.51 |
| May | 17.8 | 0.81 | 0.43 | 18.8 | 0.59 | 0.34 |
| June | 17.9 | 0.77 | 0.33 | 18.5 | 0.76 | 0.39 |
| July | 17.9 | 0.57 | 0.30 | 18.2 | 0.60 | 0.42 |
| August | 18.1 | 0.67 | 0.34 | 19.1 | 0.63 | 0.32 |
| September | 18.9 | 0.63 | 0.36 | 19.9 | 0.71 | 0.42 |
| October | 19.0 | 0.65 | 0.40 | 20.1 | 0.82 | 0.40 |
| November | 18.2 | 0.90 | 0.36 | 19.7 | 0.69 | 0.38 |
| December | 18.3 | 0.82 | 0.50 | 19.1 | 0.63 | 0.37 |

*Generation of Calibration Slopes*

In contrast to the Level-1b data from GOES-16 and GOES-17, which store digital numbers representing scaled radiances, the Level-1b from the GOES-IP visible channels store instrument counts. The counts in the GOES-R level-1b data are digital numbers or counts which are calibrated by using the GOES-R solar reflectance calibration methodology. The reflectances for GOES-R are computed by applying constant scaling factors to these counts. The GOES-IP level-1b data also contain counts, but these counts are not adjusted for the degradation of the GOES-IP visible channel detector over time. The GOES-IP visible channels were designed to maintain a dark count of 29 $C_d$ and provide a 10-bit count, C, which would have a maximum of 1023. The mean full-disk count for any one image is computed by using the simple average of the illuminated full disk, as was performed for $R_{fd}$.

$$C_{fd} = \overline{(C - C_d)}, \tag{3}$$

In addition, while the ABI level-1b output is adjusted for the sun–earth distance, the GOES-IP count values are not. To adjust them, the following formula, which is a function of the day of the year (*doy*), is used to compute the sun–earth distance factor.

$$\rho = 1 - 0.016729 \cos(0.9856(doy - 4)/180). \tag{4}$$

The calibration slope, S, is computed by using the monthly mean values of $R_{fd}$ and $C_{fd}$, using this relationship:

$$S = SBAF\rho^2 \, R_{fd} \Big/ C_{fd}. \tag{5}$$

where *SBAF* is the spectral-band adjustment factor. The spectral-band adjustment factors account for the differences in the spectral response functions of the sensors. In general, the ABI Channel 2 is narrower than the visible channels on GOES-IP (Channel 1). The SBAF values are taken from Reference [20] and are derived from convolution of the various spectral response functions with data from the Scanning Imaging Absorption Spectrometer for Atmospheric Chartography (SCHIMACHY) [21]. Table 3 provides the values used here. As this table shows, the SBAF values are very close to unity for all the GOES-IP sensors, and this adjustment does not represent a large source of uncertainty.

**Table 3.** Spectral-band adjustment factors (*SBAFs*) used in this study [20].

| Reference (*x*-Axis) | Target (*y*-Axis) | SBAF Value |
| --- | --- | --- |
| GOES-16 | GOES-8 | 1.006 |
| GOES-16 | GOES-12 | 1.011 |
| GOES-16 | GOES-13 | 0.997 |
| GOES-17 | GOES-9 | 1.005 |
| GOES-17 | GOES-10 | 1.010 |
| GOES-17 | GOES-11 | 1.013 |
| GOES-17 | GOES-15 | 0.996 |

Once *S* is generated for each month for each satellite, an equation to predict *S* as a function of time is computed. The form of the fit used for *S* is a quadratic, and it is the same format used in solar channel calibration in the Advanced Very High-Resolution Radiometer (AVHRR) by Reference [4] that is shown here.

$$S(x) = S_o\left(100 + a(x) + b(x^2)\right)/100. \tag{6}$$

The factors of 100 are there so that the linear degradation term (*a*) is expressed as %/year.

To allow the fit to account for the annual cycles in *S*, the final fit is accomplished by using Equation (2).

$$S(x) = S_o\left(100 + a(x) + b(x^2)\right) + c\sin x + d\cos x + e\sin x^2 + f\cos x^2)/100. \tag{7}$$

This equation includes harmonics of x to improve the fit performance. In the end, we assume the harmonics disappear (*c* = *d* = *e* = *f* = 0) in the applied fit, and only the terms ($S_0$, *a*, and *b*) are taken from (2) and used in (1). The resulting mean absolute difference in these slopes derived from (2) and (1) is generally about 1% of the mean slope value for each sensor. The largest deviation of 1.5% is for GOES-9, which has a very short record for fitting.

In the above equation, *x* is the time in years since the start of the calibration, $S_0$ is the calibration slope at launch (*x* = 0), and *a* and *b* are the degradation terms. The calibration start dates are shown in Table 1. Since the NESDIS operational calibration was not generated for GOES-9, the calibration start used for GOES-9 is the operational date. The fit process assumed an uncertainty in the calibration slopes given by the standard deviation of the daily $R_{fd}$ values for each month.

## 4. Results

Figure 5 shows the resulting calibration slopes derived for the GOES-IP sensors in the GOES-East position, and Figure 6 shows the calibration slopes for the GOES-IP sensors in the GOES-West position. In Figures 5 and 6, the symbols show the calibration slopes computed for each sensor and each month. The orange line (PATMOS-x) is the calibration curve derived here from these points. Table 4 provides actual coefficients of the derived fits for all GOES-IP sensors. In addition, Table 4 provides the root-mean square difference computed from the derived slopes to the individual values used in the fit. The RMS values are also called the standard errors and range in value from 2.0 to 3.5%, with a mean of 2.7%. The mean for the GOES-East is 2.0, and the mean for GOES-West is 3.3%. The largest RMS values are for GOES-9, but this sensor record is the shortest of all the GOES-IP sensors. The GOES-9 visible channel was noisy, and NOAA decided to move GOES-9 to support Japan in 1999 when the first Multifunctional Transport Satellite (MTSAT-1) had a launch failure. GOES-10 and -12 were moved to 60W to support South America after their operational service [4]. These standard errors are slightly larger than, but still similar to, those reported by Reference [4] for the calibration of the 0.65 micrometer channel on the Advanced Very

High-Resolution Radiometer (AVHRR) and similar to the standard error values reported by Reference [5].

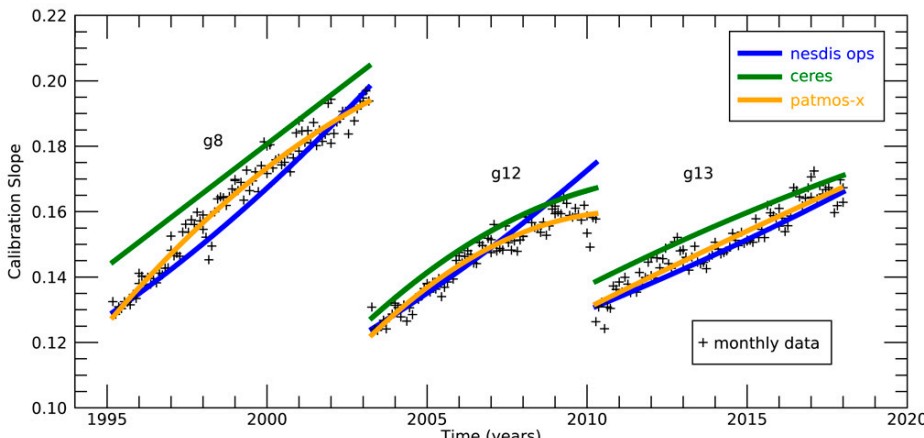

**Figure 5.** Comparison of calibration slopes. Blue line is the NESDIS operational value. Yellow line is the PATMOS-x fit derived here. Each thin black line represents the slopes computed directly for each month of the year from each satellite, which is the basis of the PATMOS-x calibration fits.

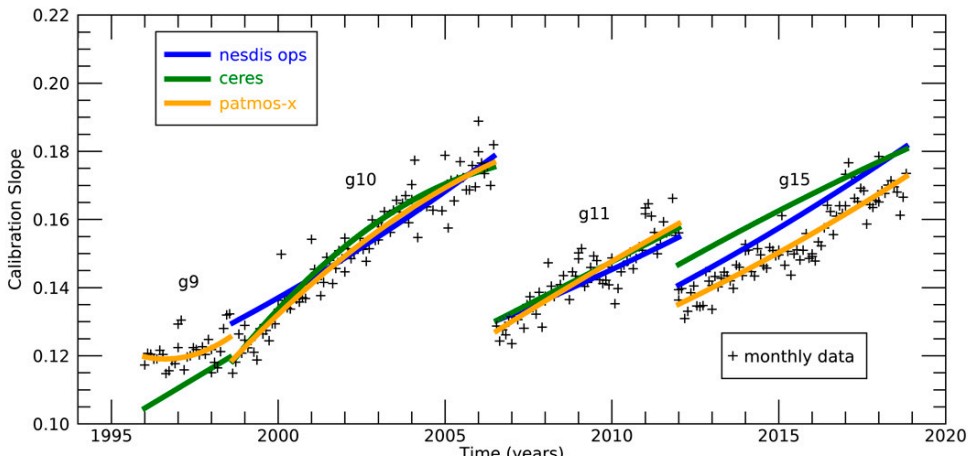

**Figure 6.** Same as Figure 5 for GOES-West.

**Table 4.** Calibration slope coefficients and metrics of the fitting performance.

| Satellite | $S_0$ | $a$ | $b$ | RMS (%) |
|-----------|-------|-----|-----|---------|
| GOES-8 | 0.130 | 8.24 | −0.250 | 2.0 |
| GOES-9 | 0.120 | −2.45 | 1.41 | 3.5 |
| GOES-10 | 0.132 | 7.02 | −0.28 | 3.2 |
| GOES-11 | 0.127 | 4.86 | −0.054 | 3.3 |
| GOES-12 | 0.122 | 7.71 | −0.473 | 1.8 |
| GOES-13 | 0.132 | 3.57 | −0.014 | 2.1 |
| GOES-15 | 0.127 | 3.40 | 0.090 | 3.1 |

Figures 5 and 6 also show the two other calibrations derived for these sensors by using other methods. The operational NOAA/NESDIS-derived calibration slopes curves for the GOES-IP imagers are shown as the blue lines. The method used by NOAA/NESDIS is an expanded version of the method given by [6], Wu's (2003) method, which minimized differences in accumulated frequency of reflectance with the NASA MODIS sensor. The GOES visible channel calibrations from the NASA Langley CERES Project are shown by the

green lines. The NASA calibration and the methodology used are given by Reference [5]. Both the NESDIS operational and NASA Langley calibration terms are converted to the same form as used here, and this information is given in Appendix A.

Table 5 provides measures of the mean bias in the calibration slopes curves derived here and those from NESDIS operations and NASA CERES expressed as a percentage relative to mean calibration slope. These values range from <1% for GOES-11 (NESDIS and NASA) to over 7% for GOES-9 (NASA) and GOES-15 (NASA). GOES-8 and GOES-9 are outside of the MODIS era, and this might explain this large difference. The cause for the large biases for GOES-15 is not obvious. GOES-9 falls entirely outside of the MODIS record and was not calibrated by the NOAA/NESDIS method. In general, this method agrees more often with the NESDIS operational calibration than the NASA CERES Ed4 calibration. Note that this paper is not trying to demonstrate if any of the methods are superior. The agreement of this simple method based on the stability of $R_{fd}$ to these two more complex methods is encouraging and motivates the use of this method for near-real-time monitoring of current visible channels. This agreement also speaks for the application of this method to GOES data prior to GOES-IP and other geostationary imagers with long records of observation of the noontime full disk from a fixed location.

**Table 5.** Comparison of calibration slopes between values derived here and those provided by NESDIS operations for each GOES-IP Satellite.

| Satellite | Relative Difference in Calibration Slopes (%) | |
| | NESDIS Operations | NASA Langley CERES Ed4 |
|---|---|---|
| GOES-8 | −2.2 | 6.1 |
| GOES-9 | NA | −7.4 |
| GOES-10 | 1.0 | 1.0 |
| GOES-11 | −0.72 | 0.48 |
| GOES-12 | 1.6 | 3.7 |
| GOES-13 | −1.4 | 3.9 |
| GOES-15 | 4.7 | 7.5 |

*4.1. Time-Series Stability*

The time series of the ratio of the monthly $R_{fd}$ values from GOES-East and GOES-West from GOES-IP divided by their GOES-R references are shown in Figures 7 and 8. Perfect agreement would be a value 1.0 for the whole record. The PATMOS-x values are constrained to hover around 1.0, with deviations coming from the monthly differences in $R_{fd}$ from the GOES-R values. The annual cycles of these deviations are also shown in Figures 3 and 4. The NESDIS operational and NASA CERES values are not constrained by the GOES-R reference values, and their deviations also include the impacts of the differences in the calibration methods. In agreement with Figures 5 and 6 and Table 5, the NESDIS operations' results agree better than NASA CERES results for GOES-East, with NASA CERES being consistently higher or brighter than NESDIS operations and PATMOS-x. Late in the life of GOES-12, NESDIS operations deviate from PATMOS-x. Early in the life of GOES-8, the NASA CERES results were roughly 10% brighter than NESDIS operations or PATMOS-x. This period also falls outside of the MODIS record, which is part of the NASA CERES calibration. For GOES-West, the bias seen in NASA CERES results are gone for GOES-10 and GOES-11. For GOES-15, a similar bias for NASA CERES as generally seen for GOES-East. For GOES-9, the PATMOS-x calibration is higher than NASA CERES, and there are no NESDIS operation values. GOES-9 also falls outside of the MODIS period, and this may impact the NASA CERES method.

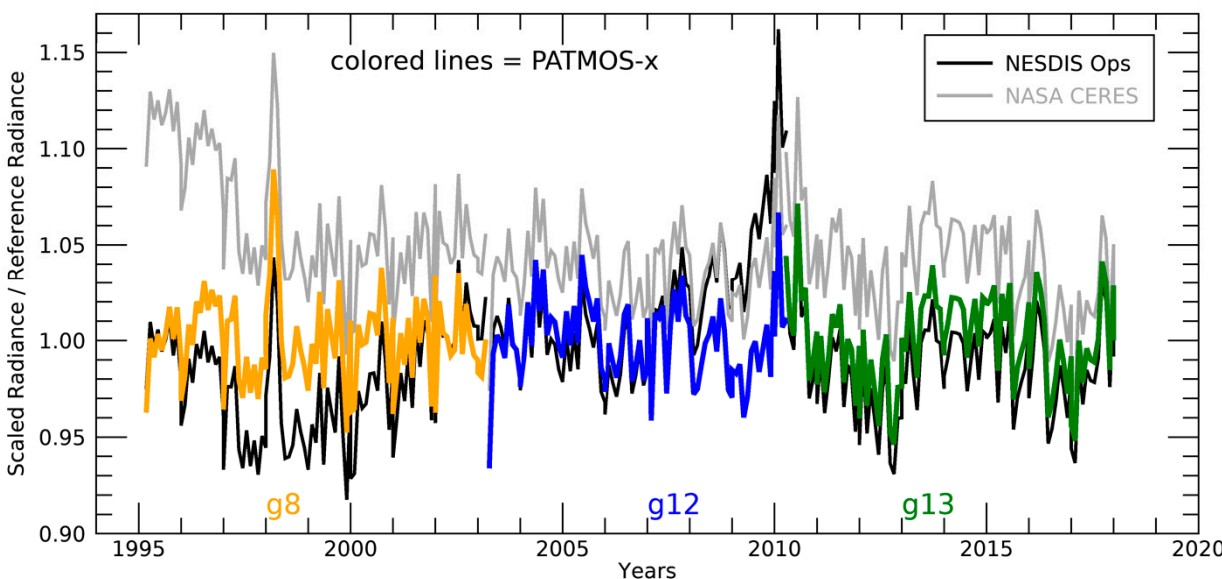

**Figure 7.** Time series of the integrated FD scaled radiance for GOES-East. Thin black line is the ABI reference. Thick colored lines are the values for each satellite from the PATMOS-x calibration derived here. Gray line shows the NESDIS operational calibration, and the black line shows the NASA CERES Ed4 calibration.

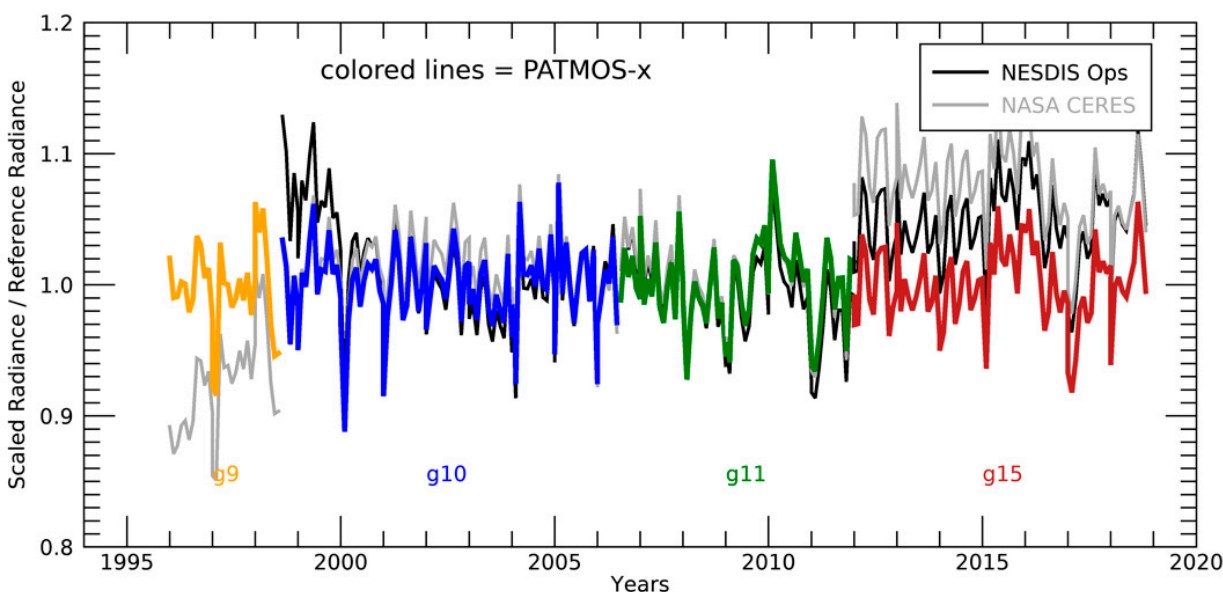

**Figure 8.** Same as Figure 7, but for GOES-West.

### 4.2. Stability of the Calibration for all Values of Reflectance

The physical basis of the calibration derived here is that the planetary albedo is stable over the GOES-West and GOES-East domains. The planetary albedo does not necessarily impose any restriction on the stability of the reflectance distributions. However, Rigollier et al. (2004) [11] derived a calibration of the Meteosat First Generation (MFG) Geostationary Imager data by effectively assuming that the reflectance distributions for noontime data are constant. Based on the success of this method, we assume that any valid calibration should also yield reflectance distributions from noontime geostationary imagers that are stable. Figures 9 and 10 show the time series of the reflectance values of three values of the cumulative distribution function (cdf) for the full-disk reflectance

distributions computed for each noontime image of GOES-IP. The data for cdf = 0.05 and 0.80 are shown since those were the values used by Reference [11], and, in addition, the values for cdf = 0.50 are shown. The planetary albedo value (basis of this calibration) corresponds approximately to cdf = 0.6. Figure 9 shows the data for GOES-East. The time series show no boundaries at the transition between GOES-8, -12, and -13. The decadal slopes that were computed by using a linear regression to data are all small numbers. The largest value of 0.84%/decade corresponds to roughly a 2% change in reflectance over 20 years. Given that the uncertainty in the calibration fits alone is 3%, these decadal slopes are not troubling.

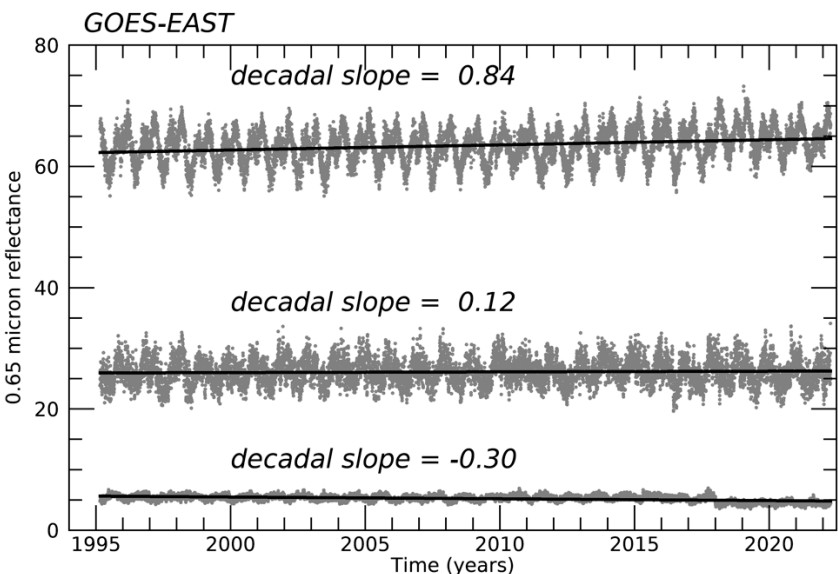

**Figure 9.** Values of 0.65 micrometer reflectance from each GOES-East full-disk image used in this study. Each series of points shows the values when the cdf curve is 0.05, 0.5, and 0.8. The linear fit to each set of points is shown as the solid lines, and the decadal slope is shown above. The units of slope are reflectance units (%) per decade.

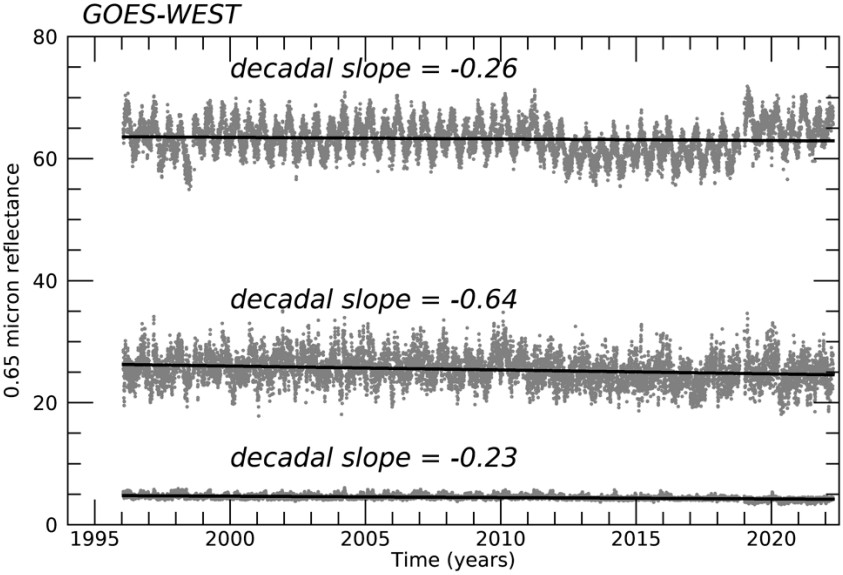

**Figure 10.** Same as Figure 9, except for GOES-West.

Figure 10 shows the same analysis for the GOES-West data (GOES-9, -10, -11, -15, and -17). One obvious difference is the large variation in the time series for GOES-West than

GOES-East. While the GOES-West time series show similar stability and magnitudes in the decadal slopes as GOES-East, there is one artifact in Figure 10. The GOES-15 results for the cdf = 0.8 show a slight drop in reflectance compared to the other satellites. This drop is not obvious in the other cdf time series. The cause of this anomaly is not known. It could be an issue in counts from the AREA files used in this analysis; however, there is no evidence for this. In exploring possible causes, the only culprit was the anomalous dark count for GOES-15. Figure 11 shows the dark or space counts computed from observing the space-viewing pixel count values in the full-disk images. The mean values are included in the legend in the figure. All GOES-IP sensors show a visible space count in the range of 29.2 to 29.4. GOES-15 is an outlier, with a mean space count of 30.6 and space count time series that shows a large variation and a slope over time. GOES-15 was unique in that it required a seasonal yaw-flip [22] because of the "dislodged thermal blanket issue", which impacted the IR calibration of the sounder. While this is suspicious and worth noting, the connection between this unique feature of GOES-15 and the issue seen in Figure 11 is not established.

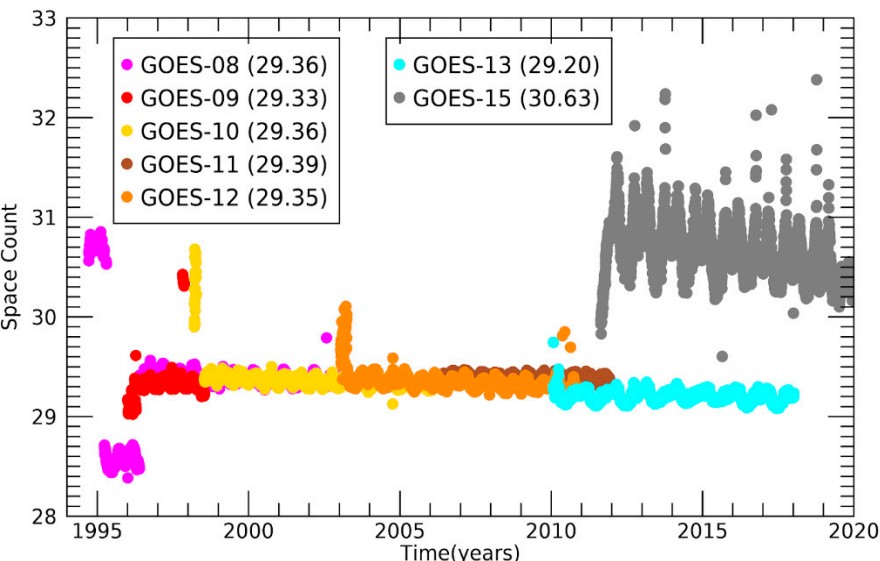

**Figure 11.** Time series of space counts from the GOES-IP sensors.

### 4.3. Verification Using Cloud Albedo over Stratus Regions

One intended application of the GOES visible calibration data is the extension of the PATMOS-x Cloud Climatology onto the GOES record. To provide additional verification of these calibration numbers, the PATMOS-x algorithms were applied to the GOES-IP and GOES-R data. One of the most sensitive PATMOS-x algorithms to the visible calibration is the Daytime Cloud Optical and Microphysical (DCOMP) algorithm [23]. The output of DCOMP includes cloud optical thickness, cloud particle size, cloud water path, cloud transmission, and cloud albedo. Here, the DCOMP output is analyzed over a region dominated by stratus clouds. Stratus clouds are composed of water droplets and are generally more spatially uniform than other cloud types. Ice clouds are not ideal for this analysis due to the uncertainty in the scattering properties of ice crystals. Stratus regions form off the west coast of continents. Given the location of the GOES-West and GOES-East sensors, only the stratus region off the coast of Baja California is visible from both at viewing angles where DCOMP operates accurately. Figure 12 shows a time series of the mean cloud albedo for a region off the coast of Baja California (24–32N, 118–124W) for July. July was chosen because it is the month where the annual cycle in the number of stratus clouds peaks in that region [23–25]. The cloud albedo was analyzed, as opposed to optical depth, since it is assumed to be proportional to reflectance, which is proportional to the calibration slopes generated here. Cloud albedo uncertainties can therefore be a surrogate for calibration slope uncertainties. The values for clear sky are assigned an albedo of zero. Figure 12 shows

that the east and west cloud albedos for the regions studied are highly correlated, with the east value typically higher than the west values. The viewing angle for the GOES-West data is 35 degrees, and for GOES-East, it is 60 degrees. Given the large difference in viewing angles, biases from the uncertainties in atmospheric modeling are expected. To translate the data in Figure 12 into a calibration verification, the mean bias between the GOES-West and GOES-East curves is assumed to be driven by algorithmic issues and is subtracted from the sensor-to-sensor biases. The mean bias between GOES-East and GOES-West is −1.8% in albedo and −7% relative to the mean of GOES-West for all years. Table 6 shows the mean albedo biases from each GOES-East/West sensor pair over the record. The biases are relative to the mean value of GOES-West for that year. As stated above, these numbers can be considered approximations of the calibration difference. These numbers generated indicate that GOES-West/East differences are less than 5%, except for the GOES-16/15, where the difference is 6.6%.

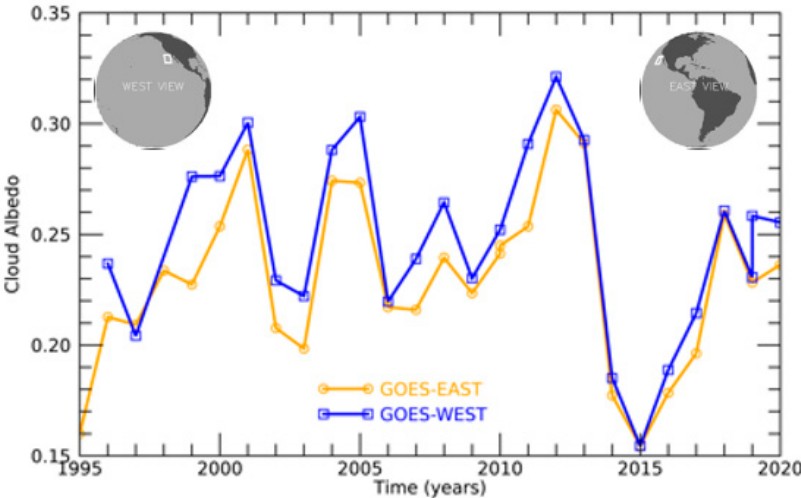

**Figure 12.** Cloud-albedo time series from GOES-East and GOES-West from GOES-IP and GOES-R. Region analyzed is 18–24N, 118–124W, and it is shown from the east and west view in the images in the upper left and right corners.

**Table 6.** Mean cloud albedo bias for the Baja stratus region shown in Figure 12 for the month of July.

| East/West Satellite Pair | Mean Albedo Bias (%) | Number of Julys in Mean |
|---|---|---|
| GOES-8/GOES-9 | 3.96 | 2 |
| GOES-8/GOES-10 | −2.95 | 4 |
| GOES-12/GOES-10 | −1.55 | 3 |
| GOES-12/GOES-11 | 1.95 | 5 |
| GOES-13/GOES-11 | −1.43 | 2 |
| GOES-13/GOES-15 | 4.20 | 6 |
| GOES-16/GOES-15 | 6.60 | 2 |
| GOES-16/GOES-17 | −2.46 | 2 |

## 5. Conclusions

The paper's main goal was to demonstrate the use of integrated full-disk visible reflectance viewed from advanced geostationary imagers to efficiently calibrate historic geostationary visible channels. This was accomplished by using the GOES-16 and GOES-17 Advanced Baseline Imager (ABI) data to recalibrate GOES-8, -9, -10, -11, -12, -13, and -15. The analysis verified that the integrated full-disk visible reflectance ($R_{fd}$) provides a stable calibration target with a monthly variability of generally less than 1% in reflectance units. Using the annual cycle of monthly $R_{fd}$ values from GOES-16 and GOES-17, we generated the derived calibration slope equations for the GOES-IP sensors with a standard error of

approximately 3%. A comparison of the derived calibration slope equations to those from other published methods shows that they generally agree within 5%. Additionally, the time series of the full-disk reflectance distributions were also found to be stable. The time-series analysis indicated an issue with GOES-15. The space counts from GOES-15 were shown to be anomalous, but it is unknown if this issue is related to the cause. In addition to these direct analyses, the time series of cloud albedo were analyzed to indicate that GOES-West and GOES-East calibrations generally agree to within 5%.

The agreement of this simple approach to other more complicated approaches motivates the application of this method to GOES data prior to GOES-IP and other geostationary imagers with long records of observation of the noontime full disk from a fixed location. Given the lack of assumptions and ease of implementation, this technique is ideal for contributing to efforts such as the Global Space-based Inter-Calibration System (GSICS), which tries to use an ensemble of techniques to generate a robust calibration information for operational geostationary imagers. This technique is also ideal for use in projects such as the International Satellite Cloud Climatology Project (ISCCP) and its follow-on versions because it directly ties the historical geostationary visible data to new records from the advanced geostationary sensors with their improved calibration and characterization. In the short term, these numbers will be used within the PATMOS-x project to generate GOES-IP cloud product climatologies to complement the existing PATMOS-x AVHRR/HIRS climatologies. These numbers will be updated as the GOES-R data records grow.

**Author Contributions:** A.K.H. conducted the analysis and conceived of the methodology with K.R.K.; T.J.S. provided knowledge on the GOES-IP sensors; M.J.F. helped review and edit the manuscript. All authors have read and agreed to the published version of the manuscript.

**Funding:** This research received no external funding.

**Data Availability Statement:** All details required to apply this calibration are included in the manuscript. All satellite data used here were provided by the Space Science and Engineering Center (SSEC) located at the University of Wisconsin–Madison.

**Acknowledgments:** The SCIAMACHY-based Spectral Band Adjustment Factors were computed from algorithms and online tools developed at NASA-LaRC with SCIAMACHY V7.01 data obtained from the European Space Agency Envisat program (Scarino et al. 2016; Bovensmann et al. 1999). The views, opinions, and findings contained in this report are those of the author(s) and should not be construed as an official National Oceanic and Atmospheric Administration or US Government position, policy, or decision.

**Conflicts of Interest:** The authors declare no conflict of interest.

## Appendix A

This appendix provides the methodology and resulting coefficients of the NESDIS operation and CERES Ed4 calibrations shown here, expressed in the same form as the PATMOS-x calibration equation.

NOAA/NESDIS provided updates of the GOES imager calibration on a monthly basis at the following site: https://www.star.nesdis.noaa.gov/smcd/spb/fwu/homepage/GOES_Imager.php (accessed on 29 May 2022).

The NOAA/NESDIS operational calibration equation for the GOES visible channels is as follows:

$$S = S_{pre}ae^{(b(y-y_c))}$$

where $y$ is time in years, and $y_c$ is the calibration start date in Table 1. $S_{pre}$ is the pre-launch calibration slope. Data from these expressions are fitted to give the parameters needed for Equation (6), and these are presented in Appendix A Table A1.

**Table A1.** NESDIS operational GOES-IP calibration expressed in the PATMOS-x format. GOES-9 was not calibrated by NESDIS operations.

| Satellite | $S_0$ | $a$ | $b$ |
|-----------|-------|-----|-----|
| GOES-8 | 0.131 | 5.27 | 0.173 |
| GOES-9 | missing | missing | missing |
| GOES-10 | 0.137 | 4.07 | 0.0972 |
| GOES-11 | 0.130 | 3.14 | 0.0558 |
| GOES-12 | 0.124 | 4.84 | 0.143 |
| GOES-13 | 0.131 | 3.06 | 0.0525 |
| GOES-15 | 0.132 | 3.71 | 0.0798 |

The NASA Langley CERES Ed4 calibration is implemented as follows:

$$S = g_o + g_1(d - d_l) + g_2(d - d_l)^2 \Big/ E_o$$

where $E_o, g_0, g_1$, and $g_2$ are given by https://www-pm.larc.nasa.gov/cgi-bin/site/showdoc?mnemonic=CALIB-ED4# (accessed on 29 May 2022).

The term $(d - d_l)$ is the time since launch and is expressed in days. NASA Langley uses the actual launch dates given in Table 1. This paper uses the calibration start dates used by NESDIS operations. Accounting for this shift in the start date, the values of $S$ versus time since calibration start are used to derive a fit of the form given in Equation (6).

**Table A2.** NASA Langley CERES Ed4 GOES-IP calibration expressed in the PATMOS-x format. The NASA Langley CERES Ed4 calibration for GOES-9 is applicable only after GOES-9 was moved out of GOES-West.

| Satellite | $S_0$ | $a$ | $b$ |
|-----------|-------|-----|-----|
| GOES-8 | 0.147 | 5.11 | 0.00 |
| GOES-9 | 0.103 | 5.69 | −0.00 |
| GOES-10 | 0.134 | 7.79 | −0.462 |
| GOES-11 | 0.130 | 3.84 | 0.00 |
| GOES-12 | 0.127 | 6.96 | −0.356 |
| GOES-13 | 0.139 | 3.52 | −0.0638 |
| GOES-15 | 0.137 | 4.20 | −0.0559 |

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
