# Peer review of "Using GOES-R ABI Full-Disk Reflectance as a Calibration Source for the GOES Imager Visible Channels"

_remotesensing, doi:10.3390/rs14153630_

Round 1

Reviewer 1 Report

I have no remarks about the text of the paper.

Author Response

I saw no specific comments to address so I simply say thank you for the review.

Reviewer 2 Report

See the attachment

Author Response

Responses to Specific Comments:

Thank you for your thoughtful comments.  My responses are in Italics.

  1. Table 1: Would you include the data start and end dates for GOES-16 and GOES-17 in this

table? That would make it easier for readers to look at the whole data record from these satellites.

Ok, I did this.  I added some text as well.  We did not re-derive GOES-16 and GOES-17 calibrations so that is why they were missing but I think your idea is a good one.

  1. Line 102-103: There were studies (e.g., NG Loeb et al. 2021) suggesting that the Earth’s energy is not in balance. Long-term trends in energy imbalance were found using satellite observations. Please comment on how this energy imbalance will affect your studies and conclusions.

I agree and I added this statement in that same area of text.

The CERES studies referenced above do indicate that the shortwave top-of-atmosphere upward flux has decreased by roughly 1W/m2 from 2000 to 2020.  Given that the mean value of this quantity is about 100 W/m2, this would represent a 1% decrease and 1% trend over 20 years in the calibration slope. In Figures 7 and 8, this would represent a change in slope on the order of one tick mark on the y-axis.  While not negligible, this violation of stability of the planetary albedo is not concern for this application.  In addition, there is strong evidence that Arctic Sea Ice has diminished during the GOES-IP era and this certainly contributes to a decrease in the planetary albedo.  The Arctic Region, however, is too far north to impact the full-disk geostationary views used here.

  1. Line 105-107: There is a strong evidence (satellite observations) that the Arctic sea ice had been shrinking over the past few decades. Didn’t this affect the planetary albedo? Please comment the effect of sea ice changes on planetary albedo or confine the statement to the areas observed by the geostationary satellites.

Yes, that is a good point and I tried to address with this statement at lines 119-122

In addition, there is strong evidence that Arctic Sea Ice has diminished during the GOES-IP era and this certainly contributes to a decrease in the planetary albedo.  The Arctic Region, however, is too far north to impact the full-disk geostationary views used here.

  1. Figure 5: Figure captions are inconsistent with Figure. Did you mean ‘black plus symbols’ for ‘black lines’?

Yes, I meant black plus symbols.  Thank you.

  1. Figures 7&8: These two figures are very important in evaluating the proposed recalibration method. Please improve the figure and consistency between captions and figures. I couldn’t see the thin black lines and dashed lines in the figures. Suggest to using thick color lines to represent the GOES-16 and GOES-17 time series. These two satellites overlap with the latest GOES-east and GOES-west satellites. It is important to understand their agreement/disagreement over the overlapping periods.

I removed the thin and dashed lines to simplify.   GOES-16 and GOES-17 are not included in this figure but I assume you means the lines based on GOES-16 and GOES-17.  Yes, they are thick colored lines.   I also did my best to change the range to provide as much visual detail as possible.

  1. Line 440: change ‘for us’ to ‘for use’

Thank you, change made.

  1. Appendix, Tables 1 and 2: Suggest to providing RMS values in both tables to compare with Table 4.

The only RMS values I could compute for these calibrations shown in the appendix are due to fitting of the PATMOS-x equation to the original functions.  These RMS values are very small and might be confusing.  The RMS values derived by the calibration providers capture the fitting of the true data and are much larger.  The NASA calibration has these numbers, but the NESDIS does not.  Given these issues, I respectfully decline this suggestion and please let me know if I misunderstood this comment.

Reviewer 3 Report

Overall, the manuscript is very well written, very interesting, and reports a new methodology developed to calibrate the visible channels on the GOES-IP sensors placed on GOES 8-13 and 15.  The authors put a lot of work into this study by analyzing a very large data set covering the years from 1994 to 2021. The manuscript is recommended to be published in its present form.

Author Response

(The authors gave the same response as above.)
